# Reconfigurable metamaterial processing units that solve arbitrary linear calculus equations

Pengyu Fu [1], Zimeng Xu [1], Tiankuang Zhou [1,2,3], Hao Li[1], Jiamin Wu [2,3] ✉, Qionghai Dai [2,3] ✉ & Yue Li [1,2] ✉

Calculus equations serve as fundamental frameworks in mathematics, enabling describing an extensive range of natural phenomena and scientific principles, such as thermodynamics and electromagnetics. Analog computing with electromagnetic waves presents an intriguing opportunity to solve calculus equations with unparalleled speed, while facing an inevitable tradeoff in computing density and equation reconfigurability. Here, we propose a reconfigurable metamaterial processing unit (MPU) that solves arbitrary linear calculus equations at a very fast speed. Subwavelength kernels based on inverse-designed pixel metamaterials are used to perform calculus operations on time-domain signals. In addition, feedback mechanisms and reconfigurable components are used to formulate and solve calculus equations with different orders and coefficients. A prototype of this MPU with a compact planar size of $0.93\lambda_0 \times 0.93\lambda_0$ ($\lambda_0$ is the free-space wavelength) is constructed and evaluated in microwave frequencies. Experimental results demonstrate the MPU's ability to successfully solve arbitrary linear calculus equations. With the merits of compactness, easy integration, reconfigurability, and reusability, the proposed MPU provides a potential route for integrated analog computing with high speed of signal processing.

A calculus equation is a mathematical expression that establishes a relationship between one or more unknown functions and their calculus, and is widely used to describe systems in economics[1], astronomy[2], geography[3], and many other disciplines[4,5]. Solving calculus equations can depict or forecast the behavior of a system. Due to the inherent complexity of most calculus equations, analytical solutions are often unattainable, making approximation through mathematical analysis the primary approach for their resolution[6–8]. Presently, these computing processes primarily rely on operational processors based on digital circuits. However, limited by transistor size, processing technology, and integration, electronic processors have difficulty achieving exponential scalability of computing density and ultra-high speed[9].

Analog computing uses electromagnetic waves as information carriers, presenting a promising avenue for performing various operations at ultra-high speed[10–13]. One of the traditional analog computing architectures is to obtain the Fourier transform with a lens and then perform various linear time-invariant operations[14]. However, these systems necessitate a spatial extent of at least four times the focal length of the lens, posing integration challenges. To address this problem, metamaterials, as artificially designed special structures, are used to realize compact analog computing units[15]. A thin planar metamaterial block is proposed to enable mathematical operations for the first time[16]. Graphene is employed in the construction of compact metalines, facilitating differentiation and integration[17]. Additionally, epsilon-near-zero metamaterials are utilized to demonstrate a

[1]Department of Electronic Engineering, Tsinghua University, Beijing, China. [2]Beijing National Research Center for Information Science and Technology, Tsinghua University, Beijing, China. [3]Department of Automation, Tsinghua University, Beijing, China. ✉e-mail: wujiamin@tsinghua.edu.cn; daiqionghai@tsinghua.edu.cn; lyee@tsinghua.edu.cn

subwavelength calculus unit and an image processing system[18]. Moreover, metastructures based on topological photonics are experimentally validated for a two-dimensional differentiation[19]. Apart from the basic mathematical operations, complex functions and tasks can be executed on large-scale systematic computing platforms, such as optical neural networks. Diverse optical neural network architectures such as diffractive deep neural networks[20–23], interference neural networks[24,25], and recurrent neural networks[26,27], have been proposed and applied to applications such as machine vision[28,29], image classification[30,31], achieving remarkable accuracy and speed. These computing platforms demonstrate computing capabilities that far exceed those of electronic devices for certain tasks, highlighting the high-speed advantages of analog computing. In addition, the extensive research on programmable and adjustable materials in various frequency bands has paved the way for the reconfigurability and multi-functionality of analog computing[32–36]. A reprogrammable plasmonic topological insulator is demonstrated for nanosecond-level state switching, which can integrate many photonic topological functionalities[34].

Solving equations based on analog computing has been widely studied as well. Over a century ago, a mechanical structure-based system for solving differential equations was proposed[37–39]. The confocal feedback system utilizing coherent optics provides a solution for solving partial calculus equations[40,41]. In order to achieve higher integration, the equation-solving system designed by optical fiber network and silicon-based technology has also been widely studied[42–44]. In addition, different types of operation units, such as memristors or topological structures, are also applied to equation solving[45,46]. In recent years, a paradigm for solving equations with inverse-designed metamaterials has been proposed. A metamaterial platform has been proposed to solve the general Fredholm integral equation of the second kind[47]. Dielectric metamaterials with different structures are used to solve differential equations in electromagnetic[48] and acoustic fields[49]. Then, an ultrathin silicon analog computing metasurface has been demonstrated to solve integral equations in free-space[50]. What is more, reconfigurable metastructures with tunable elements have been reported to solve calculus equations[51]. However, these proposals still face challenges of large sizes, reconfigurability, and compatibility, which hinder the practical application and integration of analog computing solvers with electromagnetic waves.

In this work, to overcome these limitations, we propose a reconfigurable metamaterial processing unit (MPU) for solving arbitrary calculus equations at an ultrafast speed. As shown in Fig. 1a, the MPU mainly consists of a feedback mechanism and metamaterial kernels that perform calculus operations. Distinguished from previous work in the spatial domain, we realize calculus equation solver in the time domain, which, on the one hand, allows the MPU to process continuous signals without sampling and, on the other hand, allows a significant reduction in the number of feedback mechanisms to reduce the size of the overall processing unit. Besides, the inverse-designed pixel metamaterial is used to quickly prototype kernels with calculus functions. It's worth mentioning that the metamaterial-based calculus kernels have subwavelength sizes and planar structures, thus offering the potential for integration and cascading. With reconfigurable components that consist of amplitude modulators and phase modulators, multiple differential kernels of different orders can be flexibly combined, enabling the entire system to solve arbitrary linear calculus equations. We construct a reconfigurable MPU operating in microwave frequency and experimentally verify its capability of solving linear calculus equations with arbitrary coefficients and arbitrary order. The proposed equation solvers have the advantages of compact size, integration, and reconfigurability, providing possible routes for the development of chip-based analog computers and computing elements.

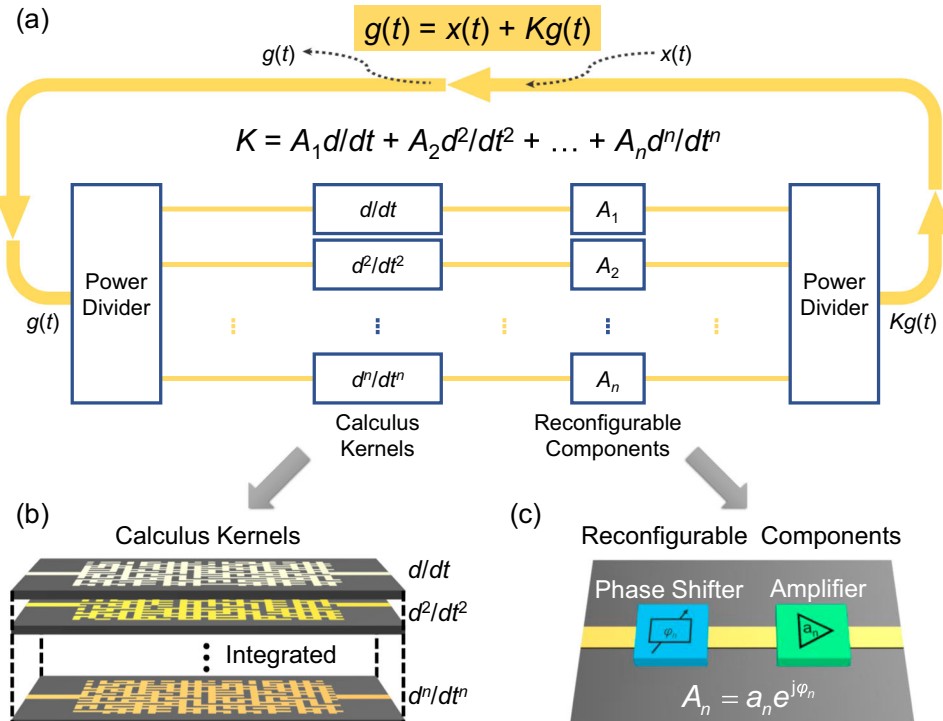

**Fig. 1 | Solving calculus equations with a reconfigurable MPU. a** Schematic representation of a reconfigurable MPU, featuring multiple processing kernels, reconfigurable elements for adjusting the amplitude and phase of each kernel, power dividers for signal composition, and a coupling element for signal excitation and probing. The arrow indicates the direction of signal propagation. **b** The different processing kernels are constructed by inverse-designed pixel metamaterials with subwavelength planar structure, and can be easily integrated into a multilayer design. **c** The reconfigurable components encompass phase shifters and amplifiers, enabling the generation of different coefficients $A_n$.

## Results

### Solving time-domain calculus equations in feedback systems

A conceptual diagram of the proposed MPU equation solver is shown in Fig. 1a. The key to building an equation solver is feedback mechanisms and analog computing kernels. Here, we consider a time-varying signal, denoted as $g(t)$, which is carried by electromagnetic waves. An inverse-designed pixel metamaterial kernel serves as a calculus operator $K$ on time domain signal $g(t)$ and the feedback mechanism directly returns the output signal from the kernel back to the input. As a result, the signals before and after operation $K$ are forced to be equal, i.e., $g(t) = Kg(t)$. Then, the input signal $x(t)$ is injected into the system through a coupler, such that the equation $g(t) = Kg(t) + x(t)$ is satisfied. This equation represents a general linear calculus equation and has a wide range of applications in thermodynamics and electromagnetism[4,5]. With the architecture depicted in Fig. 1a, we can implement arbitrary linear calculus operations. To be specific, the time-domain signal is distributed to different calculus kernels by a power divider, and the weights of each kernel can be adjusted with reconfigurable components shown in Fig. 1c, and then synthesized with a power mixer. Different types of reconfigurable components can be integrated into the system based on different operating frequencies, such as digital circuits[32], ferroelectric materials[36], or TiN microheaters[44]. Each individual calculus kernel is designed to realize calculus operations with different orders by inverse-designed pixel metamaterials, as illustrated in Fig. 1b. With a flat structure, pixel metamaterials can be easily integrated with the feedback mechanism and can be arranged into a multilayer cascading design.

For the practical implementation of MPU, considering the influence brought by the coupler, we can concretely and theoretically derive the transfer function $T$ for the whole system. We assume that the transmission coefficient and coupling coefficients of the coupler are $s_1$ and $s_2$, respectively, and that each element is reflection-free. The transfer function of the system can be defined by the following equation:

$$T(\omega) = s_1 + \frac{s_2^2 K}{1 - s_1 K} \tag{1}$$

For a conventional 3 dB hybrid coupler, the transmission coefficient is equal to the coupling coefficient with a phase delay of $\pi/2$, that is, $s_1 = js_2$. Then the above expression can be simplified to:

$$T(\omega) = \frac{s_1 - 2s_1^2 K}{1 - s_1 K} \tag{2}$$

In this case, the output signal $y(t)$ is a superposition of the signal in the loop $g(t)$ and the input signal $x(t)$, and satisfies the following calculus equation:

$$y(t) - s_1 K y(t) = s_1 x(t) - 2s_1^2 K x(t) \tag{3}$$

We use $g(t) = y(t) \cdot s_1 x(t)$ to replace the term in the calculus equation, which gives us the following expression:

$$g(t) = s_1 K g(t) - s_1^2 K x(t) \tag{4}$$

Furthermore, considering $h(t) = y(t) - 2s_1 x(t)$, the equation can be written as:

$$h(t) = s_1 K h(t) - s_1 x(t) \tag{5}$$

Based on our proposed MPU, with different calculus kernels $K$ and different input signals $x(t)$, the solutions of arbitrary linear calculus equations can be generated.

### Inverse-designed pixel metamaterial for different calculus kernels

Processing of time-domain signals often corresponds to special dispersion properties in the frequency domain. For example, an nth-order differentiating operation requires a processing kernel with the following transmit function: $T(\omega) = [j(\omega \cdot \omega_0)]^n$. We use the proposed metamaterials to achieve the optimization of the desired arbitrary computational kernels. Figure 2a illustrates the structure of the pixel metamaterial, which is composed of pixel-like patches and connecting structures. The pixel-shaped structure on the front is a metal layer, which transmits quasi-TEM (transverse electromagnetic) waves together with the bottom of the metal on the back of the dielectric plate. The main part of pixel metamaterial is composed of massive square patches with identical side length $w_p$, arranged at intervals of $d$. Two feeding lines with width $w_f$ are situated on both sides of pixel metamaterial to transmit signals. Short strips with a width $w_g$ are employed to connect adjacent patches. These strips are switched to either the ON or OFF state, where in the OFF state, the center of the line is separated by a slit with a gap of $g$, representing no connection between patches. By manipulating the states of these strips, different dispersion coefficients of the pixel metamaterials can be generated from the two feeding ports. A genetic algorithm is used in the discrete optimization process[52]. Moreover, in order to reduce the number of full-wave simulations and expedite inverse design process, we use moment method to further simplify the model of pixel metamaterials. Specifically, as shown in Fig. 2a, the ON/OFF state can be modeled as an equivalent port connected to an impedance equal to 0 or $\infty$. And then, the impedance matrix $Z_0$ of the whole system can be derived by a single-time simulation together with analytical calculations. $Z_0$ comprises the impedance matrix of two feeding ports $Z_f$, the impedance matrix of $N$ equivalent ports $Z_e$, and the impedance relationship $Z_{e,f}$, $Z_{f,e}$.

$$Z_0 = \begin{bmatrix} Z_f & Z_{f,e} \\ Z_{e,f} & Z_e \end{bmatrix} \tag{6}$$

When the ON/OFF states of the strips are determined, the impedances loaded to the $N$ equivalent ports $Z_L$ are also determined, and then the relationship between the other two freeing ports can be directly derived from the following equation:

$$Z_p = Z_f - Z_{f,e}(Z_e + Z_L)^{-1} Z_{e,f} \tag{7}$$

In this way, we can use simple matrix operations to calculate transmission parameters with different strip states. Compared with simulation-only methods, the optimization speed is greatly accelerated.

As mentioned above, in order to perform an nth-order differential kernel, the pixel metamaterials need to be optimized to fit the following transmit function: $T(\omega) = [j(\omega \cdot \omega_0)]^n$. By inverting the desired transmission parameters as an optimization objective, the calculus kernels of different orders can be obtained. Three examples of inverse-designed pixel metamaterials with first and second-order differential functions are demonstrated. Figure 2b shows the simulated results of an inverse-designed first-order differential kernel, with $n = 1$. It can be seen that at center frequency $\omega_0$, the differential kernel achieves a transmission zero point with a phase mutation of $\pi$. Within a frequency range of $-0.1\omega_0$, the amplitude of the transmission coefficient is directly proportional to the deviation from the central frequency, which meets the ideal requirements. In addition, the transmission phase changes linearly with frequency, which only brings delay to the signal without distortion. Figure 2c depicts the time-averaged surface electric-field intensity at some selected frequencies marked as squares in Fig. 2b. The energy intensity distribution at the output port also confirms that the inverse-designed kernel satisfies the transfer

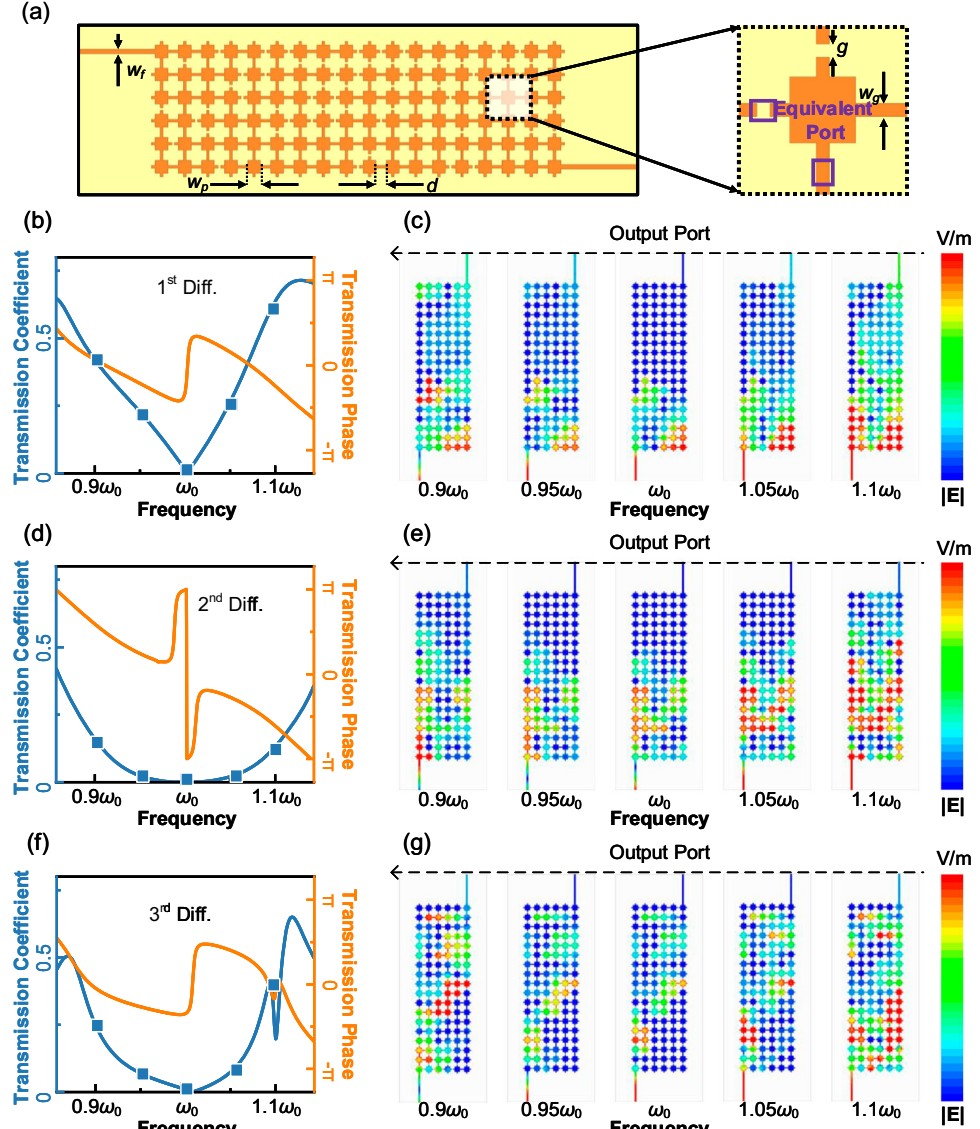

**Fig. 2 | Various differential processing kernels based on inverse design. a** Pixel metamaterial construction. The function of metamaterial is adjusted by controlling the ON/OFF state between patches. By using the moment method with equivalent ports, the time of the inverse design process can be significantly reduced. **b** Simulated results of the first-order differential kernel by inverse design. The blue line represents the amplitude of the transmission coefficient, and the orange line represents the phase of the transmission coefficient. The results accord well with the theoretical requirements of first-order differential kernel. **c** Surface electric field distribution of first-order differential kernel at different frequencies from full-wave simulation. The distribution at the output port aligns with the predicted behavior. **d** Simulated results of second-order differential kernel. **e** Surface electric field distribution of second-order differential kernel. **f** Simulated results of third-order differential kernel. **g** Surface electric field distribution of third-order differential kernel.

parameters required for first-order differentiation. Similarly, Fig. 2d–g show the simulated results of an inverse-designed second-order and third-order differential kernel, respectively, with $n = 2$, 3. The transmission coefficient is close to zero at the center frequency, with quadratic growth on both sides for the second-order kernel, and cubic growth for the third-order kernel. Differently, the phase of the second-order kernel shows a phase mutation of $2\pi$ at the center frequency, while the third-order kernel shows a phase mutation of $\pi$. To verify the differentiation function for time-domain signals, we directly process time-domain signals using the three differentiators based on inverse-designed pixel metamaterials, as shown in Fig. S2. In addition to the differential kernels implemented above, as an inverse-designed structure, pixel metamaterials hold the potential to optimize the implementation of other operation kernels, to achieve extensions of solving different types of equations.

### Experiments setup and results

Without loss of generality, we choose 1.4 GHz as the central frequency for convenient fabrication and measurements. The structural parameters used are as follows: $d = 2.4$ mm, $g = 0.6$ mm, $w_f = 1.024$ mm, $w_p = 3$ mm, $w_g = 0.6$ mm. The structure is printed on a Rogers RO4350B substrate (relative dielectric constant $\varepsilon_r = 3.66$, loss tangent $\tan\delta = 0.002$) with a thickness of 0.508 mm. The overall size of the structure is $130 \times 50$ mm$^2$, which is $0.61\lambda_0 \times 0.23\lambda_0$ ($\lambda_0$ is the free-space wavelength at the center frequency) with subwavelength footprints. By soldering sub-miniature version A (SMA) connectors at both ends of the calculus kernel, we measure the transmittance of the device using a vector network analyzer. The measurement results are in good agreement with the simulation results, as shown in Fig. S4. All these results reveal that our inverse-designed pixel metamaterial can realize arbitrary linear calculus operations within subwavelength sizes.

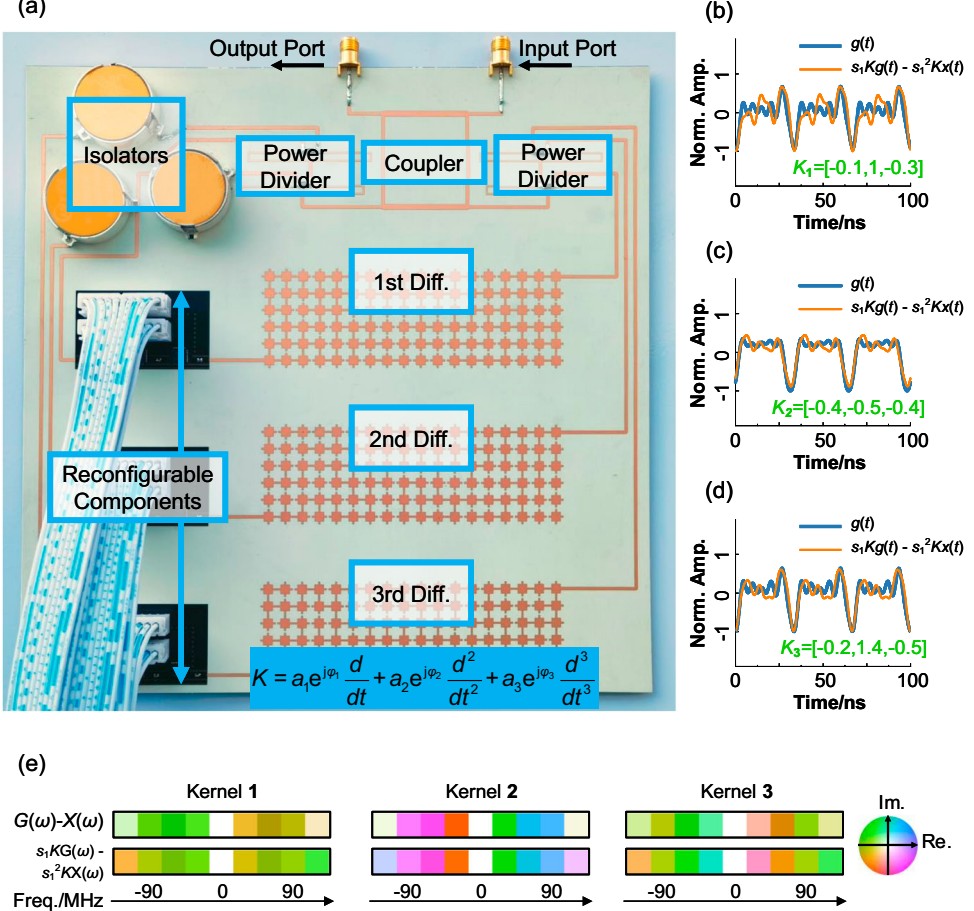

**Fig. 3 | Experimental demonstration of reconfigurable MPU that solves calculus equations. a** A photograph of the constructed reconfigurable equation solver system without DC and control circuits. This system includes a coupler, power dividers, isolators, differential kernels with different orders, as well as phase reconfigurable components controlled by FPGA. **b**–**d** Three different processing kernels are selected and demonstrated. Normalized amplitude of g(t) and $s_1Kg(t) - s_1^2Kx(t)$ with an input signal as a sawtooth wave show good consistency, respectively. **e** Spectrum comparison of G(ω) and $s_1KG(ω) - s_1^2KX(ω)$ for different processing kernels 1, 2, and 3. The left and right sides of the equation show a good agreement.

Based on the principle in Fig.1a, we construct a prototype of MPU to verify the feasibility of solving calculus equations with different calculus kernels. The photo of the experimental MPU system is shown in Fig. 3a, without DC bias and control circuits. The overall size of the experimental structure is $200 \times 200$ mm², which is ~0.93λ₀ × 0.93λ₀ with a subwavelength scale. Three different pixel metamaterial kernels are designed with first-order, second-order, and third-order differential functions, respectively. As we discussed above, the differential kernel is connected to a feedback mechanism to generate the function of solving equations. However, in the feedback loop, additional integrated components are also incorporated. The integrated isolators are included to absorb reflected signals caused by the mismatch in the feedback loop. Several compact digitally reconfigurable phase shifter chips are connected to the differential kernels, adjusting the phase delay in the loop to integer multiples of π, representing the positive and negative signs in the calculus equations. Then, compact digitally reconfigurable amplifiers with an adjustment range from −10 dB to 10 dB are used to adjust the coefficients in the calculus equations, and to ensure no self-oscillation in the MPU system. Using power dividers, the input signal x(t) is assigned to different reconfigurable calculus kernels. After synthesis as input to the feedback loop, the solution of arbitrary linear differential equations g(t) can be achieved throughout the MPU. The transmission parameters of the entire system are measured with a vector network analyzer.

The field programmable gate array (FPGA) chips are used to control all the digital phase shifters and amplifiers, with which the MPU can be switched to different states for solving different calculus equations. Without loss of generality, a triangular wave signal is used as the input signal. In the first example, we adjusted the coefficients of the three differential kernels to −0.1, 1, and −0.3. To verify the accuracy of the solution, we provide the left and right sides of the equation $g(t) = s_1Kg(t) - s_1^2Kx(t)$, respectively, as shown in Fig. 3b. The two curves match very well, indicating that the system indeed provides a solution to the calculus equation. In addition, we obtained the same consistency by setting the coefficients of the equation to [−0.4, −0.5, −0.4] and [−0.2, 1.4, −0.5] in Fig. 3c, d, respectively. Furthermore, since both input and output signals are periodic, they are discretely distributed in the frequency domain. Figure 3e shows the normalized spectra of the left and right sides of the differential equation under three different calculus kernels according to Fig. 3b–d, which are also in good agreement. Constrained by the limitations inherent in the bandwidth of the differentiators, isolators, amplifiers, and phase shifters, alongside the presence of nonlinearities within the system, it becomes evident that the solution's outcomes will inevitably exhibit a degree of error, as illustrated in Fig. 3b–d. Notably, Fig.3e underscores that the majority of this error is primarily concentrated within the high-frequency range. These errors are mainly caused by the nonlinear effects of tunable devices and the dispersion of the system. Delving

deeper into our investigation of the error-rate versus frequency relationship, the findings presented in Fig. S7 affirm that, given the prevailing experimental setup, commendable solution accuracy can be achieved when utilizing triangular waves with frequencies of up to 80 MHz. The above experiments indicate that the proposed reconfigurable architecture can solve arbitrary linear calculus equations.

At last, as shown in Table 1, a comparison is presented between our work and other existing methods for solving calculus equations. A tunable structure based on a microring resonator (MRR) is utilized to solve ordinary differential equations[44]. However, because of the large

**Table 1 | Comparison of dimension among various methods of calculus equation solver**

| Ref. | Methods | Dimension ($\lambda_O^2$) | PT (cycles) | Reconfigurable | Order extensible |
|------|---------|---------------------------|-------------|----------------|------------------|
| 44 | MRR | 38.70 × 77.40 | NG | Yes | No |
| 49 | AMS | 17.0 × 7.14 | NG | No | Yes |
| 48 | DMM | 12.0 × 16.0 | NG | No | No |
| 47 | MS | 8.63 × 4.32 | 50 | No | No |
| 50 | MG | 3.50 × 1.70 | 60 | No | No |
| Prop. | PMM | 0.93 × 0.93 | 30 | Yes | Yes |

*Ref.* reference, *Prop.* proposed scheme, *NG* not given, *MRR* microring resonators, *AMS* acoustic metasurfaces, *DMM* dielectric metamaterials, *MS* metastructures, *MG* metagratings, *PMM* pixel metamaterials, $\lambda_O$ free-space wavelength at operating frequency, *PT* processing time.

size of MRR, the size of the entire solver is greater than 30 $\lambda_0$. Besides, it is also difficult for MRR-based solvers to expand to higher orders. Accurately designed acoustic metasurfaces have been proposed for solving higher order differential equations with more integrated dimensions, but are difficult to be integrated with reconfigurable or programmable devices[49]. In addition, the idea of inverse-designed metastructures is proposed to reduce the size to a wavelength comparable level[47], about 8 $\lambda_0$. Considering a basic $3 \times 3$ units in a metagratings solving equations in free space[50], the size of the equation solver is $3.5\lambda_0 \times 1.7\lambda_0$. In contrast, as shown in Fig. 3a, due to our implementation of subwavelength differential kernels through inverse design, combined of integrated radio frequency circuits, we have achieved an MPU calculus equation solver within subwavelength scale. Meanwhile, benefiting from the integrated size, the cycles at which the input signal establishes a steady state in the MPU is significantly reduced, which means a reduction in processing time. To the best of our knowledge, this is the smallest reconfigurable solving structure for calculus equations within an ultrafast processing speed.

## Discussion

Solving calculus equations has significant applications in mechanics, electromagnetics, and other fields. Here, we present two examples of solving practical calculus equations with the proposed MPU. The first one is about earthquake-induced structural vibrations. As shown in Fig. 4a, structural vibrations occur in buildings when subjected to external forces such as earthquakes. Considering or predicting this structural vibration is an important aspect of building design. To

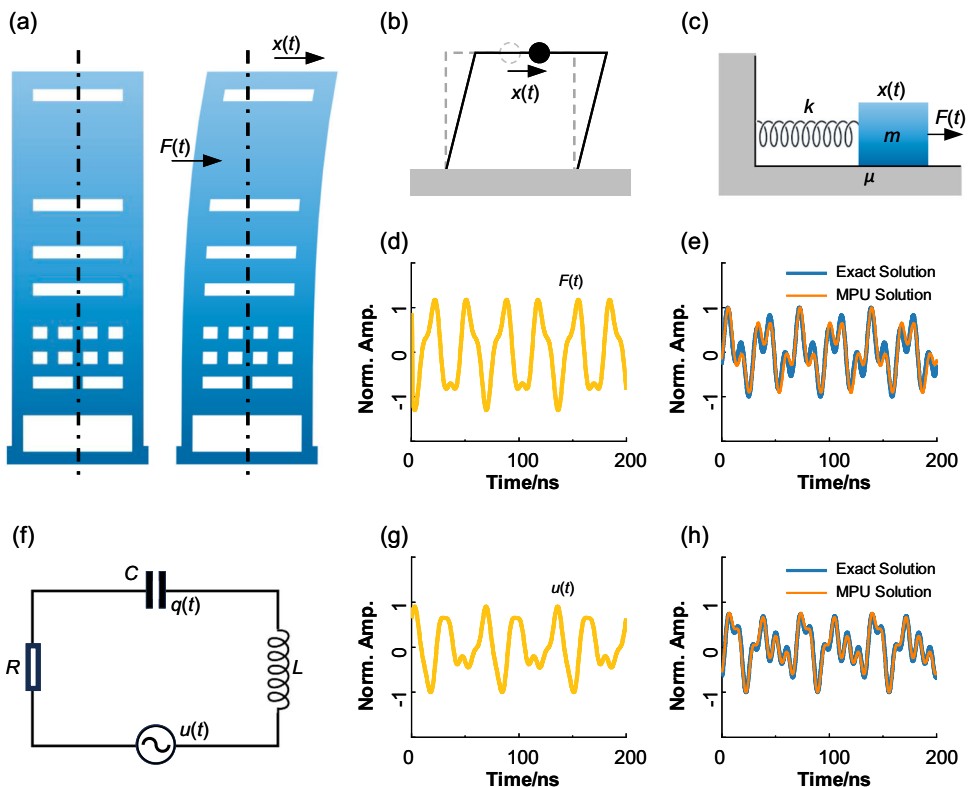

**Fig. 4 | Diagrams of practical examples for solving calculus equations. a** A schematic diagram of the vibration of a building during an earthquake, where the motion trajectory of the building is affected by external forces and damping. **b** Simplify the motion of the building to one dimension. **c** It can be modeled by a forced damped vibration model, where the applied external force F(t) and the trajectory x(t) of the object satisfy the following equation: $mx''(t) + \mu x'(t) + kx = F(t)$. **d** Assume that the external force varies as a function of time as $F(t) = \cos(2\pi t/T) + 0.3*\sin(3\pi t/T)$, where $T = 200/9$ ns. **e** It is presumed to have $m = 2$, $\mu = 0.2$, and

$k = -1.414$. The blue line is the result of numerical simulation, while the orange line is the result of MPU, which has good consistency. **f** In the RLC series circuit, a voltage $u(t)$ is applied over time, and the charge $q(t)$ at the terminals of the capacitor satisfies the equation: $Lq''(t) + Rq'(t) + q(t)/C = u(t)$. **g** Assume that the voltage varies as a function of time as $u(t) = \cos(2\pi t/T) + 0.5*\sin(3\pi t/T) + \sin(6\pi t/T)$, where $T = 200/9$ ns. **h** It is presumed to have $L = 0.2$, $R = 2$, and $C = 0.707$. The blue line is the result of numerical simulation, while the orange line is the result of MPU, which has good consistency.

simplify the problem, the structural vibration is modeled as a one-dimensional damped-forced vibration, as shown in Fig. 4b, c. Based on the principle of mechanics, the trajectory of the object $x(t)$ and the external force $F(t)$ are subjected to the following linear differential equation:

$$m\frac{d^2x(t)}{dt^2} + \mu\frac{dx(t)}{dt} + kx(t) = F(t) \tag{8}$$

The parameters of the equation are assumed as $m = 2$, $\mu = 0.2$, $k = -1.41$, and the external force varies with time satisfying the following equation: $F(t) = \cos(2\pi t/T) + 0.3*\sin(3\pi t/T)$, in which $T = 200/9$ ns, as shown in Fig. 4d. Comparing the numerical analysis solution (blue line) and the solution obtained using the MPU (orange line) in Fig. 4e, it can be observed that there is good agreement between the two results, validating the effectiveness of the MPU in solving the differential equation. The second example relates to a simple RLC series circuit, as illustrated in Fig. 4f. For a simple RLC series circuit with a time-varying voltage $u(t)$, the charge $q(t)$ between the capacitor satisfies the following equation:

$$L\frac{d^2q(t)}{dt^2} + R\frac{dq(t)}{dt} + \frac{1}{C}q(t) = u(t) \tag{9}$$

Similarly, we assume that we have the following parameters: $L = 0.2$, $R = 2$, $C = 0.71$, and $u(t) = \cos(2\pi t/T) + 0.5*\sin(3\pi t/T) + \sin(6\pi t/T)$, where $T = 200/9$ ns, as shown in Fig. 4g. Again, a comparison between the numerical simulation results and the results obtained using the MPU is depicted in Fig. 4h, demonstrating a high degree of consistency between the two. The above two examples demonstrate the practical value of our MPU for solving differential equations.

As a conclusion, we propose a reconfigurable MPU for solving arbitrary linear calculus equations in time domain with the subwavelength scale. Firstly, based on subwavelength inverse-designed pixel metamaterials, multiple kernels with different differentiation responses are designed. Then, with the FPGA, reconfigurable amplifiers, phase shifters, and power dividers are integrated with kernels to achieve arbitrary calculus operations. An experiment in the microwave band demonstrates that the equation solver can provide solutions with tiny errors for different input signals and different calculus kernels, and can be applied to practical problems in fields such as mechanics and electromagnetism. Since the calculus kernels, feedback mechanisms, and circuits that constitute the equation solver are all planar structures, the architecture has the potential for integrated multilayer design and parallel computing. Furthermore, it also holds the promise for integrated design alongside electromagnetic sensors and radio frequency devices, thereby enabling the development of specialized chips dedicated to solving calculus equations. By solving arbitrary linear calculus equations at a subwavelength scale, the proposed reconfigurable MPU offers a promising solution for next-generation analog computing systems with the merits of reconfigurability and dense integration.

## Methods
### Numerical full-wave simulations
The numerical simulations on the 3D structure of inverse-designed metamaterials have been carried out with the ANSYS HFSS 18. The copper in the model is set to the perfect electric conductor (PEC) boundary condition. Two 50-ohm lumped ports are used to excite the SMA ports in the model. As shown in Fig. 2a, we set the parameters as follows: $d = 2.4$ mm, $g = 0.6$ mm, $w_f = 1.024$ mm, $w_P = 3$ mm, $w_g = 0.6$ mm. All relative permittivity parameters taken are from the material library in the software.

### Optimization methods
The optimization of the inverse-designed metamaterials is with genetic algorithm (GA) as shown in Fig. S3. We use optimization toolbox in MATLAB 2017a in this process. The parameters of GA are: Generations = 500, PopulationSize = 500, MigrationFraction = 0.3, FitnessLimit = 0, StallGenLimit = 100. And the loss function is defined as the distance from the ideal value. The GA process runs on a personal computer equipped with an Intel (R) Core (TM) i7-10700 CPU @ 2.90 GHz and random-access memory of 64.0 GB.

### Fabrication and measurement setup
The calculus kernels are fabricated using a printed circuit board (PCB) process, with 0.508-mm thickness Rogers RO4350B dielectric with a relative dielectric constant of 3.66 and a loss tangent of 0.002. The amplifiers, phase shifters, and isolators are all commercially available integrated chips and devices. The phase shifter adopts the PE44820 chip, which is controlled by an 8-bit program and provides a 360° phase modulation function in steps of 1.4 deg near 1.4 GHz. In addition, the insertion loss of the phase shifter is about 7 dB. The amplifier adopts the QPA9126 chip, which can provide a gain of 16 dB at 1 GHz, while the PE43702 chip, which can provide attenuation of 0.25 dB to 31.75 dB by a 7-bit program in steps of 0.25 dB. Considering all the above factors, a gain adjustment range of 12.75 dB to −18.75 dB can be achieved. The isolator adopts a surface-mounted device of model UIYS125A1150T1650, providing a reverse isolation degree of 15 dB. The FPGA control circuit uses a commercial Arduino® Mega 2560 Rev3 microcontroller, integrated with an ATmega2560 processor. The vector network analyzer (KEYSIGHT FieldFox Microwave Analyzer N9951B 44 GHz) is used to measure the S-parameters of the system.

### Time-domain signals
Based on the S-parameters measured with a vector network analyzer, we use MATLAB for time-domain analysis. We conduct a discrete Fourier transform (DFT) on the input signal, then multiply it by the measured transmission parameters to obtain the frequency spectrum of the output signal, and subsequently carry out an inverse DFT to obtain the output time-domain signal. The signal we used in the experiment is a sawtooth wave with a rise time constituting 20% and modulated on a 1.4 GHz carrier.

## Data availability
The authors declare that all data needed to evaluate the conclusions in the paper are present in the paper and/or the Supplementary Materials. Additional data and codes related to this paper have been deposited in the Zenodo repository database[53] under accession code https://doi.org/10.5281/zenodo.12565807.

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

## Acknowledgements

Y.L. acknowledges partial support from the National Natural Science Foundation of China (NSFC) under grants U22B2016; J.W. acknowledges partial support from the National Natural Science Foundation of China (NSFC) under grants 62088102; And Q.D. acknowledges partial support from the National Natural Science Foundation of China (NSFC) under grants 62222508.

## Author contributions

Y.L. conceived the idea and supervised the project with J.W. and Q.D. P.F. carried out the analytical derivations and full-wave simulations and the experiments. Z.X., H.L., and T.Z. assisted in assembling the tested prototypes and constructing the experiment setup. All authors discussed the theoretical and numerical aspects, interpreted the results, and contributed to the preparation and writing of the manuscript.

## Competing interests

The authors declare no competing interests.
