## [Peer Review File · Nature Communications]

Reconfigurable metamaterial processing units that solve arbitrary linear calculus equationsREVIEWER COMMENTS

Reviewer #1 (Remarks to the Author):

Positive :

Comparison and Novelty: The paper presents a comparative analysis with existing methods for solving differential equations, highlighting the advantages of the proposed approach in terms of size reduction. The comparison emphasizes that the introduced MPU, incorporating subwavelength differential kernels through inverse design, achieves a compact size, which is a significant novelty in solving differential equations.

Thorough Technological Description: The discussion on the structure of the inverse-designed pixel metamaterial is well-explained, emphasizing its manipulation of dispersion coefficients through the control of strip states. The integration of genetic algorithms for discrete optimization and moment methods for simplified modeling showcases a comprehensive approach to achieve the desired dispersion properties. The demonstration of first and second-order differential kernel simulations effectively supports the paper's claims regarding their functionality.

Negative:

1. Impact and Comparative Evaluation: Although the paper presents a table comparing the size of different methods for calculus equation solvers, it lacks a broader comparative evaluation encompassing factors like power consumption, processing speed, or adaptability to a wider range of equations. A more comprehensive analysis of the overall impact and advantages over existing methods beyond size reduction could strengthen the conclusion's credibility.

2. Error analysis and robustness: The accuracy of simulation calculations is always an important issue to be addressed. While the examples presented show a high degree of agreement between the MPU solution and the numerical analysis, a more in-depth analysis of potential sources of error or the robustness of dealing with different types of equations and input signals would strengthen the discussion. Addressing the limitations and robustness of the system when dealing with different scenarios would provide a more comprehensive view of its practical applicability.

3. There are some grammatical mistakes throughout the text: in the annotations of Fig 2a "Adjust The function of metamaterial is adjusted by controlling the on-off state between patches". Hence, the paper needs to go undergo a thorough proofreading.

4. The definition of the reconfigurable metamaterial in this paper needs to be specified. Reconfigurable metamaterials usually modulate the refractive index of the material by electrical control. The following papers can be referred to: G. Wu, et al., Sideband-free space–time-coding metasurface antennas, *Nature Electronics*, 5, 808–819 (2022) and J. You, et al., Reprogrammable plasmonic topological insulators with ultrafast control, *Nature Communications*, 12, 5468 (2021).

Overall, the manuscript could be published pending on major revisions.

Reviewer #2 (Remarks to the Author):

The paper « Reconfigurable metamaterial processing units that solve arbitrary calculus equations » by Fu. et al. reports a reconfigurable metamaterial processor that can solve linear differential equations. A prototype working at microwave frequencies, with planar and relatively compact geometry, is designed and fabricated. Experimental results are provided.

The story of the introduction is rather standard. The claim is that to perform some operations, like solving calculus equations, all-analog systems may be advantageous as they can operate with lower power than digital ones, with possibility of parallel computations. The authors refer to the abundant literature in this field, ending their introduction with the fact that the systems previously reported still had large sizes, lacked reconfigurability and were not integrated.

I would be cautious in using the word optical with no care as in the end of the introduction, which mentions « optical analog computing solvers », since the achievement here is at microwaves where things are much easier to handle, in particular the two main claims of reconfigurability and integration. Preparation of controlled time-domain signals is also

easier at microwaves. The introduction should not oversell what the paper is actually reporting, or give false impressions.

The claim that solving occurs « at the speed of light » in these kinds of system has been recently debated, see <https://doi.org/10.1038/s42254-023-00645-5>. I would be careful with such claims. It is clear that the calculus cannot be over before the input signal has been entirely emitted and has travels several times within the system. The actual speed of computation has nothing to do with the speed of light, or even with the group velocity of the guided waves.

Is there a difference, in the minds of the authors, between «a calculus equation » and a differential equation ? The paper seems to be solving linear differential equations. Also, the claim of generality in the title is restricted right away to linear equations in the abstract, the reader feels a bit cheated by the gap between the title and the actual results.

The text of Section A should explain how « arbitrary » kernels can be encoded in the structure. Fig. 1 is actually much clearer about the fact that the Kernels are differential operators of highest order n (please give the value of n in the main text). Fig. 2 reports only first and second differentiation. Wouldn't it be more impressive to show highest order differentiations used in the processor, like $n=8$ for example ?

Fig. 3 shows a device that is still big as it is not an integrated circuit. Multilayer ICs at GHz exist, are built everyday, and I wonder why the authors didn't go for something extremely small and on a chip, since this is their main selling point. Therefore, the job seems to be only half done.

Fig. 4 shows the solving of order 2 differential linear equations using the prototype. It's nice but not very impressive. Is the third order differentiator used at all in these experiments ? Why not show a very complex example using highest derivatives ? In addition, such tasks can be solved by prior methods, so in my opinion the newest thing is the reconfigurability. Can the authors show that the system can switch from solving a given equation to another in a very short time ? That would be more impressive than the current high-school level

examples of spring mass and LC dynamics, that do not show the reconfigurability in a convincing way.

The temporal dynamics are not really used in the experiment. The authors measure the S parameters in f domain and then use this data in Matlab to simulate (digitally !) what would happen to time signals. I think they went for the easiest experiment, but for a work that claims to be first handling time signals in a reconfigurable way, and claims to get rid of digital solvers, that's a shortcoming.

Our Response to the Reviewers' Comments on our Manuscript # NCOMMS-23-51711-T entitled "Reconfigurable metamaterial processing units that solve arbitrary calculus equations"

Authors: Pengyu Fu, Zimeng Xu, Tiankuang Zhou, Hao Li, Jiamin Wu*, Qionghai Dai*, and Yue Li*

We thank the reviewers for their great efforts in reviewing our submitted manuscript. Based on these valuable comments, we have carefully carried out further study on the shortcomings of the previous design and addressed the comments point by point. The modified/added parts in the manuscript are highlighted in red.

Response to Reviewer #1:

Reviewer 1 states: "Comparison and Novelty: The paper presents a comparative analysis with existing methods for solving differential equations, highlighting the advantages of the proposed approach in terms of size reduction. The comparison emphasizes that the introduced MPU, incorporating subwavelength differential kernels through inverse design, achieves a compact size, which is a significant novelty in solving differential equations.

Thorough Technological Description: The discussion on the structure of the inverse-designed pixel metamaterial is well-explained, emphasizing its manipulation of dispersion coefficients through the control of strip states. The integration of genetic algorithms for discrete optimization and moment methods for simplified modeling showcases a comprehensive approach to achieve the desired dispersion properties. The demonstration of first and second-order differential kernel simulations effectively supports the paper's claims regarding their functionality."

Our Response: We would like to thank the reviewer for the positive comments.

Reviewer 1 states: "1. Impact and Comparative Evaluation: Although the paper presents a table comparing the size of different methods for calculus equation solvers, it lacks a broader comparative evaluation encompassing factors like power consumption, processing speed, or adaptability to a wider range of equations. A more comprehensive analysis of the overall impact and advantages over existing methods beyond size reduction could strengthen the conclusion's credibility."

Our Response: Thanks for your valuable comments. Based on your comments, we have conducted more extensive research on the performance of the metamaterial processing unit (MPU). Besides, we have compared the proposed MPU with more articles in terms of methods, dimensions, processing time, reconfigurability, and possibility of extension to higher orders, to further illustrate the innovation and advantages.

Firstly, we investigate the processing speed of the proposed MPU. For an equation solver, the processing speed can generally be determined by the time it takes for the signal to establish a steady state in the MPU [R1]. We build the corresponding simulation model in Advanced Design System® software with 2015.01 version to simulate the time-domain process for different states of the MPU, as shown in Fig. R1. The differential calculus kernels in Fig. R1(a-c) are set to only 1st order differentiation, both 1st and 2nd order differentiation, and all three different orders differentiations, respectively. Without loss of generality, an input signal with 1 GHz as the center frequency is used in the simulation. It's illustrated that for different MPU states, the output signal, which represents the solution result, reaches steady state after about 30 cycles.

Fig. R1. The simulated output signals of three different kernel states.

In addition, we investigate the power consumption of proposed MPU. The power consumption of MPUs can be mainly composed of radio frequency (RF) signals, reconfigurable components, and field programmable gate array (FPGA) control devices. For RF signals, we generate a 1mW RF signal in the vector network analyzer as the input signal for experiment. For reconfigurable components, the QPA9126 chips used as amplifiers are with the input voltage of 5V and input current of 68 mA. Meanwhile, the PE44820 and PE43713 chips, used as programmable amplitude modulators and phase shifters, have input currents of 150 μ A and 130 μ A, at an input voltage of 5V. Therefore, the total power consumption of reconfigurable components is 1024.2mW. For the FPGA devices, the Arduino Mega 2560 Rev3 microcontroller used in the experiments has an input voltage of 5V and input current of 724.3mA. All in all, the total power consumption of the proposed MPU is approximately 11889.7mW. It should be noted that the vast majority of the power consumption comes from external control and configurable devices.

We conduct an extensive comparison with other works to further illustrate the innovation and advantages as shown in Table 1. A tunable structure employing a microring resonator (MRR) has been applied in the resolution of ordinary differential equations [R2]. Nonetheless, due to the substantial dimensions of MRR, the overall size of the solver exceeds $30 \lambda_0$. Accurately designed acoustic metasurfaces that have been proposed for solving higher order differential equations feature more integrated dimensions of $17.0\lambda_0 \times 7.14\lambda_0$, but are difficult to be

integrated with reconfigurable or programmable devices [R3]. An alternative approach suggests the utilization of inverse-designed metastructures to diminish the size to a level comparable to the wavelength [R1], approximately $8 \lambda_0$. Systems utilizing dielectric metamaterials can also be implemented to solve differential equations in $12.0\lambda_0 \times 16.0\lambda_0$. Besides, employing a basic 3×3 units in metagratings to resolve equations in free space yields a solver size of $3.5\lambda_0 \times 1.7\lambda_0$ [R5]. However, these inverse-designed metamaterial schemes are difficult to integrate with reconfigurable components. As a comparison, our proposed pixel-metamaterial-based MPU has a subwavelength scale of $0.93\lambda_0 \times 0.93\lambda_0$, which is the smallest among the best known. Benefiting from this, our proposed MPU also has the shortest processing time. In addition, our proposed MPU is also with the reconfigurability and possibility of extension to higher orders.

Based on this, we modify part of Introduction for better explanation and summary of existing work:

“Solving equations based on analog computing has been widely studied as well. Over a century ago, a mechanical structure-based system for solving differential equations was proposed³⁷⁻³⁹. The confocal feedback system utilizing coherent optics provides a solution for solving partial calculus equations^{40,41}. In order to achieve higher integration, the equation solving system designed by optical fiber network and silicon-based technology has also been widely studied⁴²⁻⁴⁴. In addition, new types of operation units, such as memristors or topological structures, are also applied to equation solving^{45,46}. In recent years, a paradigm for solving equations with inverse-designed metamaterials has been proposed. A metamaterial platform has been proposed to solve the general Fredholm integral equation of the second kind⁴⁷. Dielectric metamaterials with different structures are used to solve differential equations in electromagnetic⁴⁸ and acoustic fields⁴⁹. Then, an ultrathin silicon analog computing metasurface has been demonstrated to solve integral equations in free-space⁵⁰. What is more, reconfigurable metastructures with tunable elements have been reported to solve calculus equations⁵¹. However, these proposals still face challenges of large sizes, reconfigurability and compatibility, which hinder the practical application and integration of analog computing solvers with electromagnetic waves.”

Besides, we modify part of Results at section C to strengthen the conclusion's credibility:

“At last, as shown in Table 1, a comparison is presented between our work and other existing methods for solving calculus equations. A tunable structure based on a microring resonator (MRR) is utilized to solve ordinary differential equations⁴⁴. However, because of the large size of MRR, the size of the entire solver is greater than $30 \lambda_0$. Besides, it is also difficult for MRR based solvers to expand to higher orders. Accurately designed acoustic metasurfaces have been proposed for solving higher order differential equations with more integrated dimensions, but are difficult to be integrated with reconfigurable or programmable devices⁴⁹. In addition, the idea of inverse-designed metastructures is proposed to reduce the size to a wavelength comparable level⁴⁷, about $8 \lambda_0$. Considering a basic 3×3 units in a metagratings solving equations in free space⁵⁰, the size of equation solver is $3.5\lambda_0 \times 1.7\lambda_0$. In contrast, as shown in Fig. 3(a), due to our implementation of subwavelength differential kernels through inverse design, combined of integrated radio frequency circuits, we have achieved an MPU calculus equation solver within subwavelength scale. Meanwhile, benefiting from the integrated size, the cycles at which the input signal establishes a steady state in the MPU is significantly reduced, which means a reduction in processing time. To the best of our knowledge, this is the smallest reconfigurable solving structure for calculus equations within an ultrafast processing speed.”

Ref.	Methods	Dimension (λ_0^2)	PT (cycles)	Reconfigurable	Order extensible
44	MRR	38.70×77.40	NG	Yes	No
49	AMS	17.0×7.14	NG	No	Yes
48	DMM	12.0×16.0	NG	No	No
47	MS	8.63×4.32	50	No	No
50	MG	3.50×1.70	60	No	No
Prop.	PMM	0.93×0.93	30	Yes	Yes

Table 1. Comparison of dimension among various methods of calculus equation solver.

Ref. = reference, Prop. = proposed scheme, NG = not given, MRR = microring resonators, AMS = acoustic metasurfaces, DMM = dielectric metamaterials, MS = metastructures, MG = metagratings, PMM = pixel metamaterials, λ_0 = free space wavelength at operating frequency, PT = processing time.

What's more, we add a discussion part on device power consumption in the Supplementary materials:

“Supplementary Note 7. Discussion on power consumption of MPU.

The power consumption of MPUs can be mainly composed of radio frequency (RF) signals, reconfigurable components, and field programmable gate array (FPGA) control devices. For RF signals, we generate a 1mW RF signal in the vector network analyzer as the input signal for experiment. For reconfigurable components, the QPA9126 chips used as amplifiers are with the input voltage of 5V and input current of 68 mA. Meanwhile, the PE44820 and PE43713 chips, used as programmable amplitude modulators and phase shifters, are with input currents of 150 μ A and 130 μ A, at an input voltage of 5V. Therefore, the total power consumption of reconfigurable components is 1024.2mW. For the FPGA control devices, the Arduino Mega 2560 Rev3 microcontroller used in the experiments is with input voltage of 5V and input current of 724.3mA. All in all, the total power consumption of the proposed MPU is about 11889.7mW. It should be noted that the vast majority of the power consumption comes from external control and configurable devices.”

”

[R1] [47] Mohammadi Estakhri, N., Edwards, B. & Engheta, N. Inverse-designed metastructures that solve equations. *Science* **363**, 1333-1338 (2019).

[R2] [44] Wu, J. et al. Compact tunable silicon photonic differential-equation solver for general linear time-invariant systems. *Optics express* **22**, 26254-26264 (2014).

[R3] [49] Zuo, S., Wei, Q., Tian, Y., Cheng, Y. & Liu, X. Acoustic analog computing system based on labyrinthine metasurfaces. *Scientific reports* **8**, 10103 (2018).

[R4] [48] Zhang, W., Qu, C. & Zhang, X. Solving constant-coefficient differential equations with dielectric metamaterials. *Journal of Optics* **18**, 075102 (2016).

[R5] [50] Cordaro, A. et al. Solving integral equations in free space with inverse-designed ultrathin optical metagratings. *Nature Nanotechnology*, 1-8 (2023).

Reviewer 1 states: “2. Error analysis and robustness: The accuracy of simulation

calculations is always an important issue to be addressed. While the examples presented show a high degree of agreement between the MPU solution and the numerical analysis, a more in-depth analysis of potential sources of error or the robustness of dealing with different types of equations and input signals would strengthen the discussion. Addressing the limitations and robustness of the system when dealing with different scenarios would provide a more comprehensive view of its practical applicability.”

Our Response: Thanks for your important comments. It is very important for the MPU to analyze the error and robustness of the solution. We investigate the effects of several different factors on the solution accuracy. **Firstly, the different input signals.** As shown in Fig. R2, the same MPU is used to solve equations with different input signals. It is demonstrated that quite accurate solution results for three different signals are generated, indicating the robustness of MPU with different input signals. **Secondly, the different frequency of input signals.** As shown in Fig. R3, three square waves with frequencies of 30MHz, 60MHz, and 10MHz are input into the MPU. Although solutions to the equations are generated in all three cases, it can be seen that the error in Fig. R3(b) is significantly increased. In order to quantitatively analyze the solution error, based on the content in the main text, we define the left and right sides of equations as $g(t)$ and $h(t) = s_1 \mathbf{K}g(t) - s_1^2 \mathbf{K}x(t)$, respectively. And we define the solving error in the following Eq. R1.

$$Err = \frac{\int (g(t) - h(t))^2 dt}{\sqrt{\int g(t)^2 dt \int h(t)^2 dt}} \quad (R1)$$

For the same MPU state and the same input signal waveform, we investigate the relationship between error rate of the solution and the frequency of the input signal, as shown in Fig. R4. As the frequency of the input signal increases, the error rate of the solution also increases, demonstrating that the error is caused by the bandwidth limitations of the actual physical system. **At last, the different offset of modulation frequency.** With the same definition of error rate, we analyze the relationship between modulation frequency offset and error rate of solutions. As shown in Fig. R5, as the modulation frequency deviates from the center frequency of the MPU system, the error rate increases rapidly, illustrating the importance of an accurate modulation frequency for the MPU equation solver.

Based on this, we add a discussion part on error analysis and robustness of MPU in the Supplementary materials:

“Supplementary Note 5. Error analysis and robustness of MPU.

We have investigated the effect of several different factor on the accuracy. In order to quantitatively analyze the solution error, based on the content in the main text, we define the left and right sides of equations as $g(t)$ and $h(t) = s_1 \mathbf{K}g(t) - s_1^2 \mathbf{K}x(t)$ respectively. And we define the solving error as the following Eq. S1.

$$\frac{\int (g(t) - h(t))^2 dt}{\sqrt{\int g(t)^2 dt \int h(t)^2 dt}} \quad (Eq. S1)$$

Firstly, the effect on different input signals are investigated. Here, we use the same MPU states as in Figure 3(c) in the main text. Different time domain signals with different waveform are used as input signals of the system.

As shown in Supplementary Figure 5, a simple sinusoidal signal $\cos(\omega t) + \sin(3\omega t)$, square wave signal, and triangular wave signal with the same frequency of 30MHz is used as examples. It is illuminated that quite accurate solution results for three different signals are generated, indicating the robustness of MPU with different input signals.

Secondly, the effect on the different frequencies of input signals are investigated. As shown in Supplementary Figure 6, three square waves with frequency of 30MHz, 60MHz, and 10MHz are input into the MPU. Although solutions to the equations are generated in all three cases, it can be seen that the error rate at 60MHz case is significantly increasing. Besides, as shown in Supplementary Figure 7, when the frequency of the input signal increases, the error rate of the solution grows, which demonstrates that the error is caused by the bandwidth of the actual physical system.

At last, the effect on the different offsets of modulation frequency are investigated. Predictably, since the calculus kernels require the signal to be modulated at the exact zero frequency, the frequency offset has a very large impact on the solution accuracy. As shown in Supplementary Figure 8, as the modulation frequency deviates from the center frequency of the MPU system, the error rate increases rapidly.”

Fig. R2. Supplementary Figure 5 | Solutions of MPU with different input signals. (a) The input signal is a superposition of simple sine waves. **(b)** The input signal is a square wave. **(c)** The input signal is a triangular wave.

Fig. R3. Supplementary Figure 6 | Solutions of MPU with different frequency of input signals. (a) The input signal is a square wave with frequency of 30MHz. **(b)** The input signal is a square wave with frequency of 60MHz. **(c)** The input signal is a square wave with frequency of 10MHz.

Fig. R4. Supplementary Figure 7 | Relationship between solving error rate and input signal frequency. As the frequency of the input square wave signal varies, the error rate of the solution grows, which demonstrates that the error is caused by the bandwidth of the actual physical system.

Fig. R5. Supplementary Figure 8 | Relationship between solving error rate and modulation frequency error. As the modulation frequency of the input deviates from the center frequency point, the error rate of the solution grows quickly.

Reviewer 1 states: “3. There are some grammatical mistakes throughout the text: in the annotations of Fig 2a "Adjust The function of metamaterial is adjusted by controlling the on-off state between patches". Hence, the paper needs to go undergo a thorough proofreading.”

Our Response: Thanks for your carefully reading and checking to our manuscript. We thoroughly proofread the article multiple times to make sure there are no grammatical or spelling errors in the article.

Reviewer 1 states: “4. The definition of the reconfigurable metamaterial in this paper needs to be specified. Reconfigurable metamaterials usually modulate the refractive index of the material by electrical control. The following papers can be referred to: G. Wu, et al., Sideband-free space–time–coding metasurface antennas, *Nature Electronics*, 5, 808–819 (2022) and J. You, et al., Reprogrammable plasmonic topological insulators with ultrafast control, *Nature Communications*, 12, 5468 (2021).”

Our Response: Thanks for your valuable comments. The reconfigurable materials used are further described and elaborated on. In the design of MPUs, we need to bring in reconfigurable

material to modulate and change the calculus kernel of the MPU. Reconfigurable materials, which come in different forms and operate at different frequencies, have been studied extensively. For example, digital metamaterial has the capability to manipulate EM waves in different manners by modulating the digital encoding in the metamaterial through FPGAs [R6]. Ultra-compact reconfigurable optical nano-kirigami offers an unconventional approach for realizing manipulation of nanoscale light on a chip [R7]. A reprogrammable plasmonic topological insulator is demonstrated for a nanosecond-level state switching, which can integrate many photonic topological functionalities [R8]. Moreover, space-time modulation has shown a very wide range of application scenarios in the regulation of radio frequency electromagnetic waves [R9]. In addition, ferroelectric ceramics with ion substitution design exhibits a huge range of dielectric constant adjustment in lower frequency scenarios [R10]. In the optical frequency band, TiN microheaters thermal-optical reconfigurable method has also been widely used. Considering the operating frequency band of our MPU prototype, we utilize integrated digital chips PE43713 and PE44820 as reconfigurable materials or components. The change of system state is realized by adding different control voltages on different pins of the chip. Other forms of reconfigurable materials will have potential applications when MPU prototypes with different frequencies are processed.

Based on this, we added literature research on reconfigurable material and components in Introduction section:

“In addition, the extensive research on programmable and adjustable materials in various frequency bands has paved the way for the reconfigurability and multi-functionality of analog computing³²⁻³⁶. A reprogrammable plasmonic topological insulator is demonstrated for a nanosecond-level state switching, which can integrate many photonic topological functionalities³⁴.”

And then, we added description of reconfigurable component selection in Results section:

“To be specific, the time domain signal is distributed to different calculus kernels by a power divider, and the weights of each kernel can be adjusted with reconfigurable components shown in Fig. 1(c), and then synthesized with a power mixer. Different types of reconfigurable components can be integrated in system based on different operating frequencies, such as digital circuits³², ferroelectric materials³⁶, or TiN microheaters⁴⁴.”

Besides, we add the literature [32-36] in the revised manuscript:

[R6] [32] Cui, T. J., Qi, M. Q., Wan, X., Zhao, J. & Cheng, Q. Coding metamaterials, digital metamaterials and programmable metamaterials. *Light: Science & Applications* **3**, e218-e218, doi:10.1038/lsa.2014.99 (2014).

[R7] [33] Chen, S. et al. Electromechanically reconfigurable optical nano-kirigami. *Nature Communications* **12**, 1299, doi:10.1038/s41467-021-21565-x (2021)

[R8] [34] You, J. W. et al. Reprogrammable plasmonic topological insulators with ultrafast control. *Nature communications* **12**, 5468 (2021).

[R9] [35] Wu, G.-B., Dai, J. Y., Cheng, Q., Cui, T. J. & Chan, C. H. Sideband-free space–time-coding metasurface antennas. *Nature electronics* **5**, 808-819 (2022).

[R10] [36] Li, R. et al. Giant dielectric tunability in ferroelectric ceramics with ultralow loss by ion substitution

Response to Reviewer #2:

Reviewer 2 states: “The paper « Reconfigurable metamaterial processing units that solve arbitrary calculus equations » by Fu. et al. reports a reconfigurable metamaterial processor that can solve linear differential equations. A prototype working at microwave frequencies, with planar and relatively compact geometry, is designed and fabricated. Experimental results are provided. The story of the introduction is rather standard. The claim is that to perform some operations, like solving calculus equations, all-analog systems may be advantageous as they can operate with lower power than digital ones, with possibility of parallel computations. The authors refer to the abundant literature in this field, ending they introduction with the fact that the systems previously reported still had large sizes, lacked reconfigurability and were not integrated.”

Our Response: We would like to thank the reviewer for the positive comments.

Reviewer 2 states: “1. I would be cautious in using the word optical with no care as in the end of the introduction, which mentions « optical analog computing solvers », since the achievement here is at microwaves where things are much easier to handle, in particular the two main claims of reconfigurability and integration. Preparation of controlled time-domain signal is also easier at microwaves. The introduction should not oversell what the paper is actually reporting, or give false impressions.”

Our Response: Thanks for your valuable comments. We agree that, as you mentioned, the “optical analog computing solver” may not be a very accurate overview of our proposed MPU. Wave-based analog computing is widely studied for its high processing speed and low power consumption. In particular, the form of waves carrying information can be diverse, such as acoustic waves and electromagnetic waves. In our manuscript, we focus on the reconfigurable equation solver architectures based on time-domain calculus processing kernels. Of course, as you kindly mentioned, we have developed the prototypes for experiment in the microwave band. Therefore, considering the accuracy and rigor of the manuscript, we avoid using the term “optical analog computing”, and use “analog computing” or “analog computing with electromagnetic waves” instead.

Based on this, we modify the specific wording used in several places in the manuscript:

“Analog computing with electromagnetic waves presents an intriguing opportunity to solve calculus equations with unparalleled speed, while facing an inevitable tradeoff in computing density and equation reconfigurability.”

“With the merits of compactness, easy integration, reconfigurability, and reusability, the proposed MPU provides a potential route for integrated *analog computing* with high speed of signal processing.”

“However, these proposals still face challenges of large sizes, reconfigurability and compatibility, which hinder the practical application and integration of *analog computing solvers with electromagnetic waves*.”

“The proposed equation solvers have the advantages of compact size, integration and reconfigurability, providing possible routes for the development of chip-based *analog computers and computing elements*.”

“By solving arbitrary linear calculus equations at single-wavelength scale, the proposed reconfigurable MPU offers a promising solution for next-generation *analog computing systems* with the merits of reconfigurability and dense integration.”

Reviewer 2 states: “2. The claim that solving occurs « at the speed of light » in these kinds of system has been recently debated, see <https://doi.org/10.1038/s42254-023-00645-5>. I would be careful with such claims. It is clear that the calculus cannot be over before the input signal has been entirely emitted and has travels several times within the system. The actual speed of computation has nothing to do with the speed of light, or even with the group velocity of the guided waves.”

Our Response: Thanks for your very constructive comments. Following your suggestion, we investigate the processing speed of the proposed MPU, and adjust the wording in the manuscript. For an equation solver, the processing speed can generally be determined by the time it takes for the signal to establish a steady state in the MPU [R11]. We build the corresponding simulation model in Advanced Design System® software with 2015.01 version to simulate the time-domain process for different states of the MPU, as shown in Fig. R6. The differential calculus kernels in Fig. R6(a-c) are set to only 1st order differentiation, both 1st and 2nd order differentiation, and all three different orders differentiations, respectively. Without loss of generality, an input signal with 1 GHz as the center frequency is used in the simulation. It is illustrated that for different MPU states, the output signal, which represents the solution result, reaches steady state after about 30 cycles.

Based on this comment, we have adjusted the wording in the manuscript:

“Analog computing with electromagnetic waves presents an intriguing opportunity to solve calculus equations *with unparalleled speed*, while facing an inevitable tradeoff in computing density and equation reconfigurability. Here, we propose a reconfigurable metamaterial processing unit (MPU) that solve arbitrary linear calculus equations *at a very fast speed*.”

“With the merits of compactness, easy integration, reconfigurability, and reusability, the proposed MPU provides a potential route for integrated analog computing *with high speed of signal processing*.”

“Here, to overcome these limitations, we propose a reconfigurable metamaterial processing unit (MPU) for solving

arbitrary calculus equations *at an ultra-fast speed.*”

“To the best of our knowledge, this is the smallest reconfigurable solving structure for calculus equations *within an ultrafast processing speed.*”

Besides, we have added a relevant reference into the manuscript:

[R11] [13]McMahon, P. L. The physics of optical computing. *Nature Reviews Physics* **5**, 717-734 (2023).

In addition, we add a discussion part on processing time of MPU in the Supplementary materials:

“Supplementary Note 6. Discussion on power consumption of MPU.

To evaluate the processing speed of an equation solver, it is crucial to analyze the time required for the signal to achieve a stable state within the MPU. In order to conduct this analysis, we developed a simulation model using Advanced Design System® software, version 2015.01. The simulation explores the time-domain process across various states of the MPU. Differential calculus kernels, as illustrated in **Supplementary Figure 9** (a-c), were configured to exhibit first-order differentiation, both first and second-order differentiation, and all three orders of differentiation, respectively. In the simulation, an input signal with a center frequency of 1 GHz was utilized for generality. Results indicate that regardless of the MPU's state, the output signal representing the solution stabilizes after approximately 30 cycles. This analysis provides valuable insights into the processing speed of the equation solver and highlights the effectiveness of the MPU in achieving steady state solutions within a reasonable timeframe.”

Fig. R6. Supplementary Figure 9 | The simulated output signals of three different kernel states. (a) Only 1st order differentiation. (b) Both 1st and 2nd order differentiation. (c) All three different orders differentiations.

Reviewer 2 states: “3. Is there a difference, in the minds of the authors, between «a calculus equation » and a differential equation? The paper seems to be solving linear differential equations. Also, the claim of generality in the title is restricted right away to linear equations in the abstract, the reader feels a bit cheated by the gap between the title and the actual results.”

Our Response: Thanks for your constructive comments. First of all, as you mentioned, the calculus equations we solved in the paper are general linear calculus equations, although the

reconfigurable architecture we proposed has the potential to integrate nonlinear devices. We hope to explore the problem of solving arbitrary nonlinear equations in our future work. In addition, we believe that the “calculus equations” encompass differential equations, integral equations, and hybrid equations with both calculus operations. In the manuscript, we show the solution of general linear differential equations. There are two possible ways to solve equations containing integral operations. Firstly, as we derived in Section A of manuscript, \mathbf{K} in the Eq. R2 can represent any calculus operation kernel.

$$h(t) = s_1 \mathbf{K}h(t) - s_1 x(t) \quad (\text{R2})$$

When an integral kernel is constructed and incorporated into the proposed architecture, the solution of the integral equation can be performed. However, an n^{th} order integral operation requires a processing kernel with the following transmit function: $T(\omega) = [-j/(\omega - \omega_0)]^n$, which requires an infinite response at the center frequency point. This, on the one hand, makes the approximation of kernel inaccurate, and on the other hand, it may bring unstable self-excitation to the feeding back loop of proposed architecture. In addition to this, for linear integral equations, we can transform them into linear differential equations by taking derivatives on both sides and performing variable substitution. Since our solution results in a steady-state solution of the system, we can obtain the solution of the integral equation from the results of the corresponding differential equation and the input signal. As an example, following the RLC model shown in the manuscript, the current $i(t)$ in the circuit satisfies the following calculus equation:

$$L \frac{di(t)}{dt} + Ri(t) + \frac{1}{C} \int i(t) dt = u(t) \quad (\text{R3})$$

Considering the calculus relationship between the current $i(t)$ and charge $q(t)$: $dq(t)/dt=i(t)$, we can obtain the current in the circuit by solving for the charge $q(t)$ that satisfies the differential equation.

$$L \frac{d^2 q(t)}{dt^2} + R \frac{dq(t)}{dt} + \frac{1}{C} q(t) = u(t) \quad (\text{R4})$$

In conclusion, we believe that our proposed architecture has the ability to solve arbitrary linear calculus equations. For better rigor and accuracy of the manuscript, we have avoided the use of the “arbitrary calculus equations” and used the “arbitrary linear calculus equations” instead, in title and text.

Based on this comment, we modify the title of the paper.

“Reconfigurable metamaterial processing units that solve arbitrary linear calculus equations”

Besides, we modify the specific wording used in several places in the manuscript:

“Here, we propose a reconfigurable metamaterial processing unit (MPU) that solve arbitrary linear calculus equations at a very fast speed.”

“Experimental results demonstrate the MPU's ability to successfully solve arbitrary linear calculus equations.”

“Here, to overcome these limitations, we propose a reconfigurable metamaterial processing unit (MPU) for *solving arbitrary calculus equations* at an ultra-fast speed.”

“As a conclusion, we propose a configurable MPU for *solving arbitrary linear calculus* equations in time domain with the subwavelength scale.”

Besides, we add a discussion part on integral equations in the Supplementary materials:

“Supplementary Note 8: Discussion on solving integral equations.

The “calculus equations” include differential equations, integral equations, and hybrid equations with both calculus operations. In the manuscript, we show the solution of general linear differential equations. There are two possible ways to solve equations containing integral operations. Firstly, as we derived in Section A of manuscript, \mathbf{K} in the Eq. S2 can represent any calculus operation kernel.

$$h(t) = s_1 \mathbf{K}h(t) - s_1 x(t) \quad (\text{Eq. S2})$$

When an integral kernel is constructed and integrated into the proposed architecture, the solution of the integral equation can be performed. However, an n^{th} order integral operation requires a processing kernel with the following transmit function: $T(\omega) = [-j/(\omega - \omega_0)]^n$, which requires an infinite response at the center frequency point. This, on the one hand, makes the approximation of kernel inaccurate, and on the other hand, it may bring unstable self-excitation to the feeding back loop of proposed architecture. In addition to this, for linear integral equations, we can transform them into linear differential equations by taking derivatives on both sides and variable substitution. Since our solution results in a steady-state solution of the system, we can obtain the solution of the integral equation from the results of the corresponding differential equation and the input signal. As an example, following the RLC model shown in the manuscript, the current $i(t)$ in the circuit satisfies the following calculus equation:

$$L \frac{di(t)}{dt} + Ri(t) + \frac{1}{C} \int i(t) dt = u(t) \quad (\text{Eq. S3})$$

Considering the calculus relationship between the current $i(t)$ and charge $q(t)$: $dq(t)/dt=i(t)$, we can obtain the current in the circuit by solving for the charge $q(t)$ that satisfies the differential equation.

$$L \frac{d^2 q(t)}{dt^2} + R \frac{dq(t)}{dt} + \frac{1}{C} q(t) = u(t) \quad (\text{Eq. S4})$$

Reviewer 2 states: “4. The text of Section A should explain how « arbitrary » kernels can be encoded in the structure. Fig. 1 is actually much clearer about the fact that the Kernels are differential operators of highest order n (please give the value of n in the main text). Fig. 2 reports only first and second differentiation. Wouldn't it be more impressive to show highest order differentiations used in the processor, like $n=8$ for example ?”

Our Response: Thanks for your valuable comments. The arbitrary kernels are encoded by optimization algorithm with proposed pixel metamaterials. Fig. R7(a) shows the structure of the pixel metamaterial which is composed of several patches and connecting short strips. These strips are switched to be either connected or not connected state. By manipulating the states of

these strips, different dispersion coefficients of the pixel metamaterials can be generated from the two feeding ports. As an example, for $n = 1$ as the 1st order differential kernel, the pixel metamaterials need to be optimized to fit the transmit function: $T(\omega) = j(\omega - \omega_0)$, which means a linear growth on both sides of zero point. We define the loss function associated with the target transfer function and use a genetic algorithm to minimize the loss function. Fig. R7(b) shows the simulated results of an inverse-designed first order differential kernel, with $n = 1$. It can be seen that at center frequency ω_0 , the differential kernel achieves a transmission zero point with a phase mutation of π . Within a frequency range of approximately $0.1\omega_0$, the amplitude of the transmission coefficient is directly proportional to the interpolation from the central frequency, which meets the ideal requirements. Any other arbitrary kernel is coded and optimized with a similar approach. Considering the structure of the article, we have added a note about this part in Section B.

In our experiments, we used first-, second- and third-order differential devices with $n=1,2,3$. Linear differential equations describing most physical systems have only coefficients up to third order, for example, Maxwell's equations in electromagnetism, Newton's second law in mechanics, Weibull-Holliday equation in biology, and Black-Scholes equation in economics. We use these three orders of differential devices as an example to demonstrate the accuracy of our system for solving the equations. We further show all three differential kernels used in our experiments in Fig. R7. However, for arbitrary differential kernels that are generally have more orders, in addition to optimizing them with suitable structures, we can obtain them by cascading the already obtained lower order differentiators as well. As shown in Fig. R8, we give an example of cascading to create higher order differential kernels. To eliminate reflected waves from mismatched individual differentiators, we need to add isolators or attenuators constructed from lossy media between the differentiators in the cascade. Fig. R8(b, c, d) show the results for 2nd, 3rd, and 8th order differentiators built from 1st order differential kernels and 5dB attenuators, respectively, and are in good agreement with expectations. In addition, with the planar structure of pixel metamaterials, it is also possible to construct higher-order kernels by stacking them in layers to reduce the overall size of the device.

Based on this comment, we modify the Fig. 2 and add some part about kernels encoded in Section 2.

“B. Inverse-designed pixel metamaterial for different calculus kernels

Processing of time-domain signals often corresponds to special dispersion properties in the frequency domain. For example, an n^{th} order differentiating operation requires a processing kernel with the following transmit function: $T(\omega) = [j(\omega - \omega_0)]^n$. We use the proposed metamaterials to achieve the optimization of the desired arbitrary computational kernels. Fig.2(a) illustrates the structure of the pixel metamaterial which is composed of pixel-like patches and connecting structures.”

“As mentioned above, in order to perform an n^{th} order differential kernel, the pixel metamaterials need to be optimized to fit the following transmit function: $T(\omega) = [j(\omega - \omega_0)]^n$. By inverting the desired transmission parameters as optimization objective, the calculus kernels of different orders can be obtained. Three examples of inverse-designed pixel metamaterials with first and second order differential functions are demonstrated. Fig. 2(b)

shows the simulated results of an inverse-designed first order differential kernel, with $n = 1$.”

“Similarly, Fig. 2(d-e) and Fig. 2(f-g) show the simulated results of an inverse-designed second-order and third-order differential kernel respectively, with $n = 2, 3$. The transmission coefficient is close to zero at the center point with quadratic grows on both sides for second-order kernel, and cubic grows for third-order kernel. Differently, the phase of the second-order kernel shows a phase mutation of 2π at the center frequency point, while the third-order kernel shows a phase mutation of π .”

Fig. R7 Fig.2 Various differential processing kernels based on inverse design.

(a) Pixel metamaterial construction. The function of metamaterial is adjusted by controlling the on-off state between patches. By using momentum method with equivalent ports, the time of inverse design process can be significantly reduced. (b) Simulated results of first-order differential kernel by inverse design. The blue line represents the amplitude of transmission coefficient, and orange line represents the phase of transmission coefficient. The results accord well with the theoretical requirements of first-order differential kernel. (c) Surface electric field distribution of first-order differential kernel at different frequencies from full-wave simulation. The distribution at the output port aligns with the predicted behavior. (d) Simulated results of second-order differential kernel. (e) Surface electric field distribution of second order differential kernel. (f) Simulated results of third-order differential kernel. (g) Surface electric field distribution of third order differential kernel.

Besides, we add a discussion of higher order extensions in the supplementary material.

Fig. R8 Supplementary Figure 10 | Higher order expansion of equation solver. By cascading differential kernels and attenuators, higher order differential kernels can be realized.

“Supplementary Note 9: Discussion on higher order expansion of equation solver.

For arbitrary calculus kernels that are generally have more orders, in addition to optimizing them using suitable pixel metamaterial structures, we can obtain them by cascading the already obtained lower order differentiators as well. As shown in **Supplementary Figure 10**, we give an example of cascading to create higher order differential kernels. To eliminate reflected waves from mismatched individual differentiators, we need to add isolators or attenuators constructed from lossy media between the differentiators in the cascade. **Supplementary Figure 10** (b, c, d) show the results for 2nd, 3rd, and 8th order differentiators built from 1st order differential kernels and 5dB attenuators, respectively, and are in good agreement with expectations. In addition, with the planar structure of pixel metamaterials, it is also possible to construct higher-order kernels by stacking them in layers to reduce the overall size of the device.”

Reviewer 2 states: “5. Fig. 3 shows a device that is still big as it is not an integrated circuit. Multilayer ICs at GHz exist, are built every day, and I wonder why the authors didn’t go for something extremely small and on a chip, since this is their main selling point. Therefore, the job seems to be only half done.”

Our Response: Thanks for your important comments. Based on your suggestion, we have rebuilt a more complete prototype by integrating the coupler, isolators, power dividers, tunable elements, and the calculus kernels into the same dielectric board to achieve a further reduction in the overall size. The photo of the experimental MPU system is shown in Fig. R9(a), without DC bias and control circuits. The structure is printed on a Rogers RO4350B substrate (relative dielectric constant $\epsilon_r = 3.66$, loss tangent $\tan \delta = 0.002$) with a thickness of 0.508 mm. The overall size of the experimental structure is $200 \times 200 \text{ mm}^2$, which is approximately $0.93\lambda_0 \times 0.93\lambda_0$ *with a subwavelength scale*.

In addition, we must state that the main size of the whole system is dedicated to the calculus kernel operating time-domain signal processing. For a wave-based analog computing device, we need to allow the electromagnetic wave to propagate through the system for a sufficient length to accomplish the processing of the information. For example, A metallic waveguide system utilizing an inverse design structure to solve the equations occupies an area of 8.6×4.3 wavelengths [R12]. Whereas the equation solver based on a non-uniform dielectric medium requires 16 wavelengths to propagate the electromagnetic signal [R13]. Systems designed in acoustics that can solve differential equations also require 17 wavelengths [R14]. Even performing simple mathematical operations often requires wavelength comparable scales. As an example, linear convolution operations based on metasurfaces require about 10×12 wavelengths [R15]. In fact, to the best of our knowledge, the subwavelength scale MPU we proposed is already the smallest structure for solving calculus equations, as shown in the following Table R1.

Ref.	Methods	Dimension (λ_0^2)
[R12]	Metastructures	8.63×4.32
[R13]	Dielectric metamaterials	12.0×16.0
[R14]	Acoustic metasurfaces	17.0×7.14
Prop.	Pixel metamaterials	0.93×0.93

Table R1. Comparison of dimension among various methods of calculus equation solver.

[R12] Mohammadi Estakhri, N., Edwards, B. & Engheta, N. Inverse-designed metastructures that solve equations. *Science* **363**, 1333-1338 (2019).

[R13] Zhang, W., Qu, C. & Zhang, X. Solving constant-coefficient differential equations with dielectric metamaterials. *Journal of Optics* **18**, 075102 (2016).

[R14] Zuo, S., Wei, Q., Tian, Y., Cheng, Y. & Liu, X. Acoustic analog computing system based on labyrinthine metasurfaces. *Scientific reports* **8**, 10103 (2018).

[R15] Silva, A. et al. Performing mathematical operations with metamaterials. *Science* **343**, 160-163 (2014).

Based on this comment, we modify the Fig. 3 and add some part about experiment prototype in Section 3.

“C. Experiments setup and results

Without losing generality, we choose 1.4 GHz as the central frequency for convenient fabrication and measurements. The structural parameters used are as follows: $d = 2.4$ mm, $g = 0.6$ mm, $w_f = 1.024$ mm, $w_p = 3$ mm, $w_g = 0.6$ mm. The structure is printed on a Rogers RO4350B substrate (relative dielectric constant $\epsilon_r = 3.66$, loss tangent $\tan \delta = 0.002$) with a thickness of 0.508 mm. The overall size of the structure is 130×50 mm², which is $0.61\lambda_0 \times 0.23\lambda_0$ (λ_0 is the free-space wavelength at the center frequency) with subwavelength footprints.”

“Based on the principle in Fig.1(a), we conduct a prototype of MPU to verify the feasibility of solving calculus equations with different calculus kernels. The photo of the experimental MPU system is shown in Fig. 3(a), without DC bias and control circuits. The overall size of the experimental structure is 200×200 mm², which is

approximately $0.93\lambda_0 \times 0.93\lambda_0$ with a subwavelength scale.”

“The integrated isolators are included to absorb reflected signals caused by mismatch in the feedback loop. Several compact digitally reconfigurable phase shifter chips are connected to the differential kernels, adjusting the phase delay in the loop to integer multiples of π , representing the positive and negative signs in the calculus equations. Then, compact digitally reconfigurable amplifiers with an adjustment range from -10dB to 10dB are used to adjust the coefficients in the calculus equations, and to ensure no self-oscillation in MPU system.”

Fig. R9 Fig.3 Experimental demonstration of reconfigurable MPU that solves calculus equations.

(a) A photograph of the constructed reconfigurable equation solver system without DC and control circuits. This system includes coupler, power dividers, isolators, differential kernels with different orders, as well as phase reconfigurable components controlled by FPGA. (b-d) Three different processing kernels are selected and generated. Normalized amplitude of $g(t)$ and $s_1 K g(t) - s_1^2 K x(t)$ with input signal as sawtooth wave show good consistency respectively. (e) Spectrum comparison of $G(\omega)$ and $s_1 K G(\omega) - s_1^2 K X(\omega)$ with different processing kernel 1&2&3. The left and right sides of equation show a good agreement.

Reviewer 2 states: “6. Fig. 4 shows the solving of order 2 differential linear equations using the prototype. It’s nice but not very impressive. Is the third order differentiator used at all in these experiments ? Why not show a very complex example using highest derivatives ? In addition, such tasks can be solved by prior methods, so in my opinion the newest thing is the reconfigurability. Can the authors show that the system can switch from solving a given equation to another in a very short time? That would be more impressive than the current high-school level examples of spring mass and LC dynamics, that do not show the reconfigurability in a convincing way.”

Our Response: Thanks for your constructive comments. In Fig. 4 of the manuscript, we demonstrate that solving two simple equations with practical physical significance. Here, we mainly wish to illustrate to researchers from various domains and backgrounds the potential application value of our proposed MPU in practical scenarios. As depicted in the figure, the actual physical scenarios of kinematics and electromagnetism that we use can be accurately described by second order linear differential equations. Therefore, we deactivated the amplifier for the third order differential kernel in these two examples and use only the first and second order differential kernels. Furthermore, examples of solving more complex calculus equations using all three proposed differential kernels are given in Fig. R10.

As your constructive suggestion, it’s very important to emphasize the reconfigurability of our proposed architecture. Unlike previous work where the amplifier and phase shifter devices were separated from each other, the integrated MPU is designed with FPGAs to control all the reconfigurable components in the system. Photographs of the FPGA, the control circuitry, and the entire experimental environment can be found in Fig. R11. The phase shifter uses the PE44820 chip, which is controlled by an 8-bit program and provides a 360° phase modulation function in steps of 1.4 deg near 1.4GHz. The amplifier uses the QPA9126 chip, which can provide a gain of 16dB at 1GHz, while PE43702 chip, which can provide attenuation of 0.25dB to 31.75dB by a 7-bit program in steps of 0.25dB. The FPGA control circuit employs a commercial Arduino® Mega 2560 Rev3 microcontroller, which is integrated with ATmega2560 processor. As shown in Fig. R10, we use the FPGA to control the MPU to switch among multiple predefined states for the accurate solution of different calculus equations.

In addition, we further investigate the switching time of the system. The delay of the MPU to switch among multiple states is mainly limited by several factors: the time required for the system to establish a steady-state solution, the switching time of the reconfigurable components, and the clock signal of the FPGAs. We build the corresponding simulation model in Advanced Design System® software with the 2015.01 version to simulate the time-domain process for different states of the MPU. Results indicate that regardless of the MPU's state, the output signal representing the solution stabilizes after approximately 30 cycles, implying a settling time of about 21ns. In addition, the chips used for amplitude modulation and phase shifter have a settling time of 1600ns and 365ns respectively. Furthermore, the FPGA chip used in the experiments has a main frequency of 16MHz, implying a switching time of up to 62.5ns. Taken together, the switching time of the MPU is mainly limited by the settling time of the reconfigurable chips, which limits the whole MPU prototype to switch different solver state as

fast as 25kHz. It should be noted that different reconfigurable components and materials are key to the switching speed of the MPU system.

Based on this comment, we redesign the MPU prototype and conduct experiments, and modify the experiment setup in Section 3 with Fig.3 .

“The field programmable gate array (FPGA) chips are to control all the digital phase shifters and amplifiers, with which the MPU can be switched to different states for solving different calculus equations. Without loss of generality, a triangular wave signal is used as input signals. In the first example, we adjusted the coefficients of the three differential kernels to -0.1, 1, and -0.3. To verify the accuracy of the solution, we provide the left and right sides of the equation $g(t) = s_1 K g(t) - s_1^2 K x(t)$, respectively, as shown in Fig. 3(b).”

Fig. R10 Fig.3 Experimental demonstration of reconfigurable MPU that solves calculus equations.

(a) A photograph of the constructed reconfigurable equation solver system without DC and control circuits. This system includes coupler, power dividers, isolators, differential kernels with different orders, as well as phase reconfigurable components controlled by FPGA. (b-d) Three different processing kernels are selected and generated. Normalized amplitude of $g(t)$ and $s_1 K g(t) - s_1^2 K x(t)$ with input signal as sawtooth wave show good consistency respectively. (e) Spectrum comparison of $G(\omega)$ and $s_1 K G(\omega) - s_1^2 K X(\omega)$ with different processing kernel 1&2&3. The left and right sides of equation show a good agreement.

Besides, we modify the Materials and Methods sections.

“Fabrication and measurement setup

The calculus kernels are made with printed circuit board (PCB) process, with 0.508-mm Rogers RO4350B dielectric with relative dielectric constant 3.66 and loss tangent 0.002. The amplifiers, phase shifters, and isolators are all used commercial integrated chips and devices. The phase shifter adopts the PE44820 chip, which is controlled by an 8-bit program and provides a 360° phase modulation function in steps of 1.4 deg near 1.4GHz. In addition, the insertion loss of the phase shifter is about 7dB. The amplifier adopts the QPA9126 chip, which can provide a gain of 16dB at 1GHz, while PE43702 chip, which can provide attenuation of 0.25dB to 31.75dB by a 7-bit program in steps of 0.25dB. Considering all the above factors, a gain adjustment range of 12.75dB to -18.75dB can be achieved. The isolator adopts a surface pasting device of model UIYS125A1150T1650, providing a reverse isolation degree of 15dB. The FPGA control circuit uses a commercial Arduino® Mega 2560 Rev3 microcontroller, integrated with ATmega2560 processor.”

In addition, we add a discussion part on FPGA control scheme and switching time.

“Supplementary Note 10: Discussion on FPGA control scheme and switching time.

The **Supplementary Figure 11** gives the principle of controlling the switching state of the system with FPGA. A commercial Arduino® Mega 2560 Rev3 microcontroller integrated with ATmega2560 processor is used as the controller. As shown in **Supplementary Figure 11 (a)**, this type of FPGA chips has multiple input and output ports. And as in **Figure 11 (c)**, the pins and feeding ports of the two chips used as amplifiers and phase shifters in the experiment are connected to the output ports of the FPGA.

As for the switching time, the delay of the MPU to switch among multiple states is mainly limited by several factors: the time required for the system to establish a steady-state solution, the switching time of the reconfigurable components, and the clock signal of the FPGAs. As we discuss in the **Supplementary Note 6**, the output signal representing the solution stabilizes after approximately 30 cycles, implying a settling time of about 21ns. In addition, the chips used for amplitude modulation and phase shifter have a settling time of 1600ns and 365ns respectively. Furthermore, the FPGA chip used in the experiments has a main frequency of 16MHz, implying a switching time of up to 62.5ns. Taken together, the switching time of the MPU is mainly limited by the settling time of the reconfigurable chips, which limits the whole MPU prototype to switch different solver state as fast as 25kHz. It should be noted that different reconfigurable components and materials are key to the switching speed of the MPU system.”

Fig. R11 Supplementary Figure 11 | FPGA control scheme. (a) Commercial Arduino® Mega 2560 Rev3 microcontroller, integrated with ATmega2560 processor. (b) The FPGA is connected to the MPU via a row of wires. (c) Control circuit for MPU reconfigurable chips.

Reviewer 2 states: “7. The temporal dynamics are not really used in the experiment. The authors measure the S parameters in f domain and then use this data in Matlab to simulate (digitally !) what would happen to time signals. I think they went for the easiest experiment, but for a work that claims to be first handling time signals in a reconfigurable way, and claims to get rid of digital solvers, that’s a shortcoming.”

Our Response: Thanks for your valuable comments. We have processed and tested the prototype of proposed MPU architecture at microwave frequency to illustrate the innovations of compact size and reconfigurability. As you mentioned, our approach uses S-parameters obtained via a vector network analyzer to perform time-domain analysis via MATLAB. We start the process by subjecting the input signal to a discrete Fourier transform (DFT), followed by multiplication with the measured transmission parameters to derive the frequency spectrum of the output signal. A reverse DFT operation is then performed to generate the time-domain output signal. We believe that the simple methods we use are reasonable and accurate. The same measurements of S-parameters are used in many of the analog computation studies to illustrate the functionality of the system. For example, in the inverse-designed metastructures, multiple ports of the equation solver are sequentially excited to obtain the S-parameters, which are then subjected to synthesis of arbitrary input signals [R16]. For the optical metagratings that solve integral equations, incident lights at different angles are sequentially excited to obtain the S-parameter response and are used as an orthogonal basis to synthesize arbitrary input signals [R17]. However, in future work, we hope to be able to directly integrate our proposed MPUs with sensors or photodetectors in order to completely escape the digital computing constraints. Once again, we appreciate your thoughtful review and constructive comments.

[R16] Mohammadi Estakhri, N., Edwards, B. & Engheta, N. Inverse-designed metastructures that solve equations. *Science* **363**, 1333-1338 (2019).

[R17] Cordaro, A. et al. Solving integral equations in free space with inverse-designed ultrathin optical metagratings. *Nature Nanotechnology*, 1-8 (2023).

REVIEWERS' COMMENTS

Reviewer #1 (Remarks to the Author):

The authors have addressed all my comments and suggestions. I think the paper is ready to move to the next stage toward publication.

Reviewer #2 (Remarks to the Author):

The authors performed some significant extra work to address the comments of both referees, and I believe that the manuscript now showcase a better adequacy between claims and actual data.

Our Response to the Reviewers' Comments on our Manuscript # NCOMMS-23-51711-A entitled "Reconfigurable metamaterial processing units that solve arbitrary linear calculus equations"

Authors: Pengyu Fu, Zimeng Xu, Tiankuang Zhou, Hao Li, Jiamin Wu*, Qionghai Dai*, and Yue Li*

We sincerely thank reviewers for their great efforts in reviewing our submitted manuscript. Benefiting from these very valuable suggestions, we believe that the innovation and rigor of the paper has been enhanced. Below, we provide separate point-by-point response to the reviewers' comments.

Response to Reviewer #1:

Reviewer 1 states: "The authors have addressed all my comments and suggestions. I think the paper is ready to move to the next stage toward publication."

Our Response: We appreciate the reviewer for the significant effort and positive comments.

Response to Reviewer #2:

Reviewer 2 states: "The authors performed some significant extra work to address the comments of both referees, and I believe that the manuscript now showcase a better adequacy between claims and actual data."

Our Response: We thank the reviewer for carefully reviewing our manuscript and providing these positive comments and recommendation.